# Pioglitazone Ameliorates Lipopolysaccharide-Induced Behavioral Impairment, Brain Inflammation, White Matter Injury and Mitochondrial Dysfunction in Neonatal Rats

**DOI:** 10.3390/ijms22126306

**Published:** 2021-06-11

**Authors:** Jiann-Horng Yeh, Kuo-Ching Wang, Asuka Kaizaki, Jonathan W. Lee, Han-Chi Wei, Michelle A. Tucci, Norma B. Ojeda, Lir-Wan Fan, Lu-Tai Tien

**Affiliations:** 1Department of Neurobiology, Shin Kong Wu Ho-Su Memorial Hospital, Taipei 111, Taiwan; m001074@ms.skh.org.tw; 2School of Medicine, Fu Jen Catholic University, Xinzhuang Dist., New Taipei City 24205, Taiwan; wkc200@gmail.com (K.-C.W.); ae7750@gmail.com (H.-C.W.); 3Department of Anesthesiology, Shin Kong Wu Ho-Su Memorial Hospital, Taipei 111, Taiwan; 4Department of Pharmacology, Toxicology and Therapeutics, Division of Toxicology, School of Pharmacy, Showa University, Shingawa-ku, Tokyo 142-8555, Japan; asuka.0110@pharm.showa-u.ac.jp; 5Department of Pediatrics, Division of Newborn Medicine, University of Mississippi Medical Center, Jackson, MS 39216, USA; jlee4@umc.edu (J.W.L.); nojeda@umc.edu (N.B.O.); lwfan@umc.edu (L.-W.F.); 6Department of Anesthesiology, University of Mississippi Medical Center, Jackson, MS 39216, USA; mtucci@umc.edu

**Keywords:** pioglitazone, lipopolysaccharide, interleukin-1β, microglia activation, lipid peroxidation, mitochondrial activity

## Abstract

Previous studies have demonstrated that pioglitazone, a peroxisome proliferator-activated receptor gamma (PPARγ) agonist, inhibits ischemia-induced brain injury. The present study was conducted to examine whether pioglitazone can reduce impairment of behavioral deficits mediated by inflammatory-induced brain white matter injury in neonatal rats. Intraperitoneal (i.p.) injection of lipopolysaccharide (LPS, 2 mg/kg) was administered to Sprague–Dawley rat pups on postnatal day 5 (P5), and i.p. administration of pioglitazone (20 mg/kg) or vehicle was performed 5 min after LPS injection. Sensorimotor behavioral tests were performed 24 h after LPS exposure, and changes in biochemistry of the brain was examined after these tests. The results show that systemic LPS exposure resulted in impaired sensorimotor behavioral performance, reduction of oligodendrocytes and mitochondrial activity, and increases in lipid peroxidation and brain inflammation, as indicated by the increment of interleukin-1β (IL-1β) levels and number of activated microglia in the neonatal rat brain. Pioglitazone treatment significantly improved LPS-induced neurobehavioral and physiological disturbances including the loss of body weight, hypothermia, righting reflex, wire-hanging maneuver, negative geotaxis, and hind-limb suspension in neonatal rats. The neuroprotective effect of pioglitazone against the loss of oligodendrocytes and mitochondrial activity was associated with attenuation of LPS-induced increment of thiobarbituric acid reactive substances (TBARS) content, IL-1β levels and number of activated microglia in neonatal rats. Our results show that pioglitazone prevents neurobehavioral disturbances induced by systemic LPS exposure in neonatal rats, and its neuroprotective effects are associated with its impact on microglial activation, IL-1β induction, lipid peroxidation, oligodendrocyte production and mitochondrial activity.

## 1. Introduction

Increasing evidence indicates that perinatal infection or inflammation and hypoxia-ischemia are major contributors to white matter disease and result in the subsequent development of impaired neurological outcomes [1,2]. The treatment of white matter diseases in newborns is urgently needed to further reduce the incidence of neurological diseases in adulthood. In previous studies, we found that systemic injection of an endotoxin, lipopolysaccharide (LPS), is responsible for the majority of inflammatory effects [3], which results in white matter injury and induction of pro-inflammatory cytokines including interleukin-1β (IL-1β), interleukin-6 (IL-6) and tumor necrosis factor-alpha (TNF-α) concentrations in P5 neonatal rats [4,5]. Treatments aimed at inhibiting pro-inflammation factors might provide a protection against white matter disease.

Pioglitazone, a peroxisome proliferator-activated receptor gamma (PPAR-γ) agonist, has exhibited anti-inflammatory effects [6,7] and has been shown to prevent acute liver injury induced by ethanol and LPS through the suppression of TNF-α production in rats [8]. In addition, pioglitazone partially reduces a secondary inflammatory response in the ischemic insult especially in the endothelial and perivascular tissues of rats [9]. Activation of the PPAR-γ receptor by pioglitazone protects against neuronal and cognitive degeneration elicited by binge alcohol exposure linked to inhibition of pro-inflammatory cytokines in rats [10]. In addition, pioglitazone treatment greatly attenuates microglial activation and improved dopaminergic neuronal survival in the nigrostriatal system of rats after diffuse brain injury [11,12]. Thus, pioglitazone may potentially protect against brain injury induced by infection or inflammation via attenuating the activation of microglia and production of cytokines in neonates. However, the effect of pioglitazone on neurobehavioral disturbances induced by brain injury resulting from pro-inflammatory factors in neonates has not yet been studied.

In the present study, we investigated whether pioglitazone could reduce neurobehavioral disturbances that are associated with increased cytokine levels and reduction of mitochondrial activity induced by systemic LPS treatment in the neonatal rat brain.

## 2. Results

### 2.1. Pioglitazone Attenuated LPS-Induced Hypothermia and Body Weight Loss

The average rectal temperature is around 34 °C in control pups, and hypothermia was defined as rectal temperature of less than 32 °C [13]. The rectal temperature dropped in the LPS-treated group from 2 to 6 h after LPS injection as compared to the control group (*p* < 0.05, Figure 1A). Pioglitazone treatment significantly reduced LPS-induced hypothermia from 2 to 6 h after LPS injection (*p* < 0.05, Figure 1A). Systemic LPS injection in P5 rats resulted in a lower body weight at P6 as compared with the control group (*p* < 0.05, Figure 1B). Pioglitazone treatment significantly improved LPS-induced weight reduction and neuromuscular deficits in rats (*p* < 0.05, Figure 1B). No gender differences in LPS-induced hypothermia and lower body weigh were observed in the rats.

### 2.2. Pioglitazone Improved LPS-Induced Neurobehavioral Deficits

Compared with the saline injection, LPS injection in P5 rats resulted in sensorimotor behavioral deficits at P6 (Figure 2 and Figure 3). No sex differences in LPS-induced sensorimotor behavioral dysfunction were observed in the rats. Pioglitazone treatment significantly improved sensorimotor behavioral performance following LPS exposure (Figure 2 and Figure 3).

*Hind-Limb Suspension:* This test is used to evaluate the proximal hind-limb muscle strength, weakness and fatigue in rat neonates [14]. The LPS group exhibited significant reductions of mean latency times as compared with the control group at P6 (*p* < 0.05, Figure 2A). The reduction in hind-limb suspension latency was reversed after treatment with pioglitazone (*p* < 0.05, Figure 2A).

*Wire-hanging maneuver test:* On P6, the mean latency times of the LPS-injected group were significantly less than the control group (*p* < 0.05, Figure 2B). The reduction in wire-hanging latency times was much less prominent in the LPS + pioglitazone group than in the LPS group (*p* < 0.05), and there was no difference in wire-hanging maneuver latency times between the control and LPS + pioglitazone groups (Figure 2B).

*Negative Geotaxis:* Negative geotaxis is used to test reflex development, motor skills and vestibular labyrinth and cerebellar integration. As shown in Figure 3A, the LPS group exhibited significantly longer mean latency times for negative geotaxis along a 15° incline compared with the control group at P6 (*p* < 0.05, Figure 3A). The elongation of negative geotaxis latency in the saline + vehicle or saline + pioglitazone group was much less prominent than in the LPS group (*p* < 0.05, Figure 3A). The elongation of negative geotaxis latency was reversed after treatment with pioglitazone in the LPS group.

*Righting reflex:* As shown in Figure 3B, on P6, significantly longer mean latency times were seen in the LPS-injected group compared with the control group (*p* < 0.05, Figure 3B). Pioglitazone treatment significantly shortened the LPS-induced increase in righting reflex latency in P6 rats (*p* < 0.05, Figure 3B).

### 2.3. Pioglitazone Attenuated LPS-Induced Loss of Oligodendrocytes

Abundant late oligodendrocyte progenitor cells (O4+/O1−), which had positive staining in the cell membrane and processes, were found in the cingulum white matter of the brain sections at the bregma level in P6 control rat brains (Figure 4A–L, red). DAPI was used simultaneously to identify nuclei in the final visualization (blue) (Figure 4I–L). LPS injection reduced the number of normal O4+ cells (*p* < 0.05, Figure 4C,G,K,Q). Pioglitazone attenuated neonatal LPS-induced reduction of O4+ cells in the P6 rat brain (*p* < 0.05, Figure 4D,H,L,Q).

The total oligodendrocytes were characterized by oligodendrocyte lineage transcription factor 2 (Olig2+). Olig2+ cells were widespread in the white matter of P6 rat brain (Figure 4M–P, red). LPS injection reduced the number of Olig2+ cells (*p* < 0.05, Figure 4O,R), and pioglitazone reduced neonatal LPS-induced reduction of Olig2+ cells in the P6 rat brain (*p* < 0.05, Figure 4P,R). The ratio of late oligodendrocyte progenitor cells (O4+/O1−) to total oligodendrocytes (Olig2+) in each group were shown in Figure 4S.

Mature oligodendrocytes were identified with the APC-CC1 antibody. APC-CC1+ cells were primarily observed in the cingulum (Figure 5A–D,M–P, green), the corpus callosum and the subcortical white matter tract and in the internal capsule area at the bregma level. Same sections were also labeled with Olig2+ staining (Figure 5E–L, red). As described previously, the total oligodendrocytes were characterized by oligodendrocyte lineage transcription factor 2 (Olig2+). Double-labeling (yellow) showed that APC-CC1+ cells were Olig2+ cells (Figure 5I–L, yellow).

LPS exposure significantly reduced the number of APC-CC1+ cells (*p* < 0.05, Figure 5C,K,O,Q) and Olig2+ cells (*p* < 0.05, Figure 5G,K,R) compared with the saline or saline + pioglitazone rat brain. Pioglitazone treatment attenuated the LPS-induced reduction in mature oligodendrocytes (*p* < 0.05, Figure 5D,L,P,Q) and total oligodendrocytes in the P6 rat brain (*p* < 0.05, Figure 5H,L,R). The ratio of mature oligodendrocytes (APC-CC1+) to total oligodendrocytes (Olig2+) in each group were shown in Figure 5S.

### 2.4. Pioglitazone Attenuated LPS-Induced Reduction of Mitochondrial Complex I Activity

Mitochondrial complex I activity was measured as the amount of NADH oxidized per minute per milligram of protein in whole brain homogenates 24 h after LPS injection. Similar to our previous study [15,16], systemic LPS exposure reduced enzymatic activity of mitochondrial complex I after 24 h (*p* < 0.05, Figure 6). Pioglitazone treatment attenuated the LPS-induced decrease in mitochondrial complex I activity in P6 rat brains (*p* < 0.05, Figure 6).

### 2.5. Pioglitazone Reduced the LPS-Induced Increases in Thiobarbituric Acid-Reactive Substances (TBARS) Content

To investigate the effects of LPS and/or pioglitazone administration on oxidative stress, lipid peroxidation was assessed in brain tissues by measuring thiobarbituric acid-reactive substances (TBARS), as previously established [17,18,19]. The levels of TBARS in brain samples from the LPS group was remarkably elevated compared with the control group (*p* < 0.05) twenty-four hours following LPS injection (Figure 7). Pioglitazone treatment reduced LPS-induced increases in the levels of TBARS in the brains of P6 rats (*p* < 0.05) (Figure 7).

### 2.6. Pioglitazone Decreased the LPS-Induced Increase in Microglial Activation and Inflammatory Responses

In the control rat brain, a few microglia in the rat brain as indicated by Iba1 immunostaining were detected and most of those cells were in resting status with a ramified shape (Figure 8A,E,I). LPS treatment caused activation of microglia in the rat brain as indicated by the significantly increased numbers of activated microglia showing bright staining of an elongated or a round-shaped cell body with blunt or no processes found in the white matter in cingulum area of the rat brain 24 h after LPS injection (*p* < 0.05, Figure 8C,G,K,M). In response to LPS challenge, not only the number of Iba1+ microglia increases, but also the soma of these cells become larger. Therefore, Iba1 staining was also quantified by measuring the percentage area that contains Iba1 immunostaining in the captured images. A higher percentage of Iba1 immunostaining area was observed in the cingulum area of white matter in the neonatal LPS-exposed rat brain (*p* < 0.05, Figure 8N). Pioglitazone treatment reduced the number of activated microglia (*p* < 0.05, Figure 8D,H,L,M) and percentage of Iba1 immunostaining area (*p* < 0.05, Figure 8N) following LPS exposure.

Three major proinflammatory cytokines, IL-1β, IL-6 and TNF-α, were determined in the serum and brain 24 h after LPS exposure in P5 rats (Figure 9). Concentrations of IL-1β in the serum and brain of LPS-exposed rats were significantly increased compared to those in control rats at 24 h (*p* < 0.05, Figure 9A,B). Pioglitazone attenuated the LPS-induced increase of IL-1β in both serum and brain (*p* < 0.05, Figure 9A,B). IL-6 and TNF-α concentrations in the LPS-exposed rat serum were also significantly increased as compared to those in the control rats 24 h following LPS injection (*p* < 0.05, Figure 9A). Pioglitazone treatment reduced the LPS-induced increase of IL-6 and TNF-α levels in the serum of P6 rats (*p* < 0.05, Figure 9A).

## 3. Discussion

Consistent with our previous study, systemic LPS exposure causes loss of oligodendrocytes and mitochondrial activity, and the increase in IL-1β levels and microglia activation in the developing brains of neonatal rats [15,20]. In the present study, PPAR-γ agonist pioglitazone treatment significantly improved LPS-induced neurobehavioral disturbances including the loss of body weight, hypothermia, hind-limb suspension, wire-hanging maneuver, negative geotaxis, and righting reflex in neonatal rats (Figure 1, Figure 2 and Figure 3). The neuroprotective effect of pioglitazone against the loss of oligodendrocytes and mitochondrial activity is associated with attenuation of LPS-induced increases in IL1-β levels, the number of activated microglia, and lipid peroxidation in neonatal rat brains. Pioglitazone treatment reduces neonatal systemic LPS-induced hypothermia (<32.0 °C), which may be caused by LPS-promoted vasodilation, decreased huddling behavior, increased heat losses and reduced thermogenesis [13]. Our previous study showed that cyclooxygenase-2 (COX-2) inhibitor celecoxib protects against LPS-induced hypothermia [20] and may be associated with the inhibition of COX-2-induced hypotension [21]. Thus, the protective effects of pioglitazone against LPS-induced hypothermia may possibly result from the inhibition of LPS-induced COX-2 expression and PGE2 production, which interferes with NFκB and JNK signaling pathways [22,23].

In the current study, significant body weight loss in the rat was observed 24 h after LPS injection (Figure 2B). LPS-induced growth failure may be caused by the combination of protein loss, increased substrate utilization, reduced intake, dehydration, and diarrhea with associated fluid loss [24,25]. LPS administration resulted in poor performances in the hind-limb suspension, wire hanging maneuver, negative geotaxis, and righting reflex (Figure 2 and Figure 3). Performance in these behavioral tests is related closely to the general condition of animals [25,26]. It is possible that the poor performance in the LPS group was associated with their weaker strength due to lower body weight and dehydration which can greatly affect these behavioral results, and in such conditions these tests may not reflect neurobehavioral deficits correctly. It has been reported that body weight loss resulting from TNF-α-induced activation of ubiquitin–proteosome system-mediated muscle proteolysis [25,26,27]. LPS-induced IL-1β may selectively inhibit one type of thirst when it is produced by systemic maneuvers that produce selective dehydration on body fluid compartments [24,28]. Pioglitazone treatment not only improved neurobehavioral deficits induced by LPS (Figure 2 and Figure 3), but also reduced LPS-induced elevation of IL-1β and TNF-α levels in the neonatal rat (Figure 9). This observation provides supportive evidence for the possible involvement of muscle weakness and dehydration in the LPS-induced neurobehavioral deficits. The possible contributions of muscle proteolysis and dehydration to the poor neurobehavioral performance in the LPS group should not be excluded. Our additional experiments have further demonstrated that the protective effects of pioglitazone on LPS-induced brain injury and by improved neurobehavioral performances in juvenile rats (L-W. Fan, unpublished observations).

PPARγ is expressed on cells of the central nervous system including microglia, astrocytes, oligodendrocytes, and neurons [22,29]. In the present study, P5 neonatal LPS insult causes a reduction in total oligodendrocytes (Olig2+) (Figure 4 and Figure 5), late oligodendrocyte progenitor cells (O4+) (Figure 4) and mature oligodendrocytes (APC-CC1+) in the white matter cingulum area (Figure 5), which may contribute to hypomyelination and associated sensorimotor deficits in rats (Figure 2 and Figure 3). Oligodendrocytes are myelin-producing cells in the central nervous system, and loss of oligodendrocytes can lead to hypomyelination, which compromises nerve conduction and leads to neural network dysfunctions related to sensorimotor regulation [30,31]. Treatment with pioglitazone reduced neonatal LPS-induced reduction in both late oligodendrocyte progenitor cells and mature oligodendrocytes (Figure 4 and Figure 5), and also improved sensorimotor neurobehavioral performance in LPS-treated neonatal rats (Figure 2 and Figure 3). It has been suggested that pioglitazone protects against white matter damage by inhibiting microglial activation and promoting oligodendroglial maturation, and that these effects appear to be PPARγ-receptor dependent [22,32]. PPARγ plays a role in oligodendrocyte maturation during early postnatal development by increasing myelin basic protein expression, elaborated cholesterol-enriched membranes, peroxisomes and by promoting oligodendrocyte processes formation [22,32,33,34].

In addition, oligodendrocyte differentiation is highly dependent on mitochondrial activity, which is often affected by microglia-induced pro-inflammatory cytokines [34,35]. It has been reported that pro-inflammatory cytokines inhibit oligodendrocyte differentiation by affecting mitochondrial activity, which is associated with increased mitochondrial superoxide production, decreased mitochondrial membrane potential (mMP), and decreased ADP-induced Ca^2+^ oscillations [34,35]. Results from the present study show that pioglitazone prevents LPS-induced reductions in mitochondrial complex I activity in the neonatal rat brain (Figure 6). Hoffmann et al. have proposed that mitochondria are particularly apt to mediate brain programming by early-life stress and to serve at the same time as subcellular substrate in the programming process [36]. Pioglitazone is a known PPARɣ agonist, and Corona and Duchen [37] have demonstrated that PPARɣ is a potential target to rescue mitochondrial function in neurological disease. Thus, the protection of mitochondria activity by pioglitazone may contribute to the protective effects against LPS-induced neonatal neurobehavioral dysfunction. It has also been reported that pioglitazone reduces LPS-induced mitochondrial dysfunction and neurodegeneration in adult rats [7]. Other studies indicated that PPARγ-receptor agonist protected oligodendrocyte progenitors from TNF-α and rotenone-induced mitochondrial activity inhibition by increasing the expression of PGC-1α (a mitochondrial biogenesis master regulator), UCP2 (a mitochondrial protein known to reduce ROS production), and cytochrome oxidase subunit COX1 [34,35]. Therefore, pioglitazone may not only promote oligodendrocyte differentiation but also activate mitochondrial cell survival signaling to prevent oligodendrocyte injury, suggesting further studies are needed.

Our results show that pioglitazone prevents LPS-induced lipid peroxidation by the reduction of TBARS contents in neonatal rat brains (Figure 7). Currently, other studies also have indicated that pioglitazone treatment reduced TBARS levels in dementia animal models [38,39]. Pioglitazone, a PPARɣ agonist, has been shown to have anti-inflammatory effects [11,12]. PPARγ is a ligand-activated nuclear receptor transcription factor that regulates the function and expression of complex gene networks and may directly modulate the expression of several antioxidant and prooxidant genes in response to oxidative stress [32,40]. The PPARγ agonist promotes antioxidant defenses by increasing the expression of catalase and copper-zinc superoxide dismutase while maintaining the overall homeostasis of the glutathione system [32,40]. Moreover, Polvani et al. have reported that PPARγ agonist enhances the expression of manganese SOD (MnSOD), GPx3, the scavenger receptor CD36, endothelial nitric oxide synthase (eNOS), HO-1, and the mitochondrial uncoupling protein 2 (UCP2), whereas it downregulates COX-2 and iNOS [32,40].

PPARγ has been demonstrated to modulate inflammatory responses, including those in the central nervous system [22,29]. Our current results show that treatment with PPAR-γ agonist pioglitazone reduces LPS-induced increases of the number of activated microglia and pro-inflammatory cytokine IL-1β levels in the neonatal rat brain (Figure 8 and Figure 9). Activation of PPAR-γ promotes anti-inflammatory effects as a negative regulator of macrophage and microglia activation to reduce expression of pro-inflammatory mediators such as cytokines and chemokines [6,7,9]. PPARγ activation directly restricts tissue injury by inhibiting the NFκB pathway to reduce inflammation and stimulating the Nrf2/ARE axis to neutralize oxidative stress [29]. PPARγ serves as a master gatekeeper of cytoprotective stress responses, improving the chances of cellular survival and recovery of homeostatic equilibrium in central nervous system injury and repair [29]. In addition, several studies reported that pioglitazone decreases the production of pro-inflammatory cytokines and inhibits the expression of pro-inflammatory enzyme iNOS and reactive oxygen species (ROS) in models of LPS-induced liver or lung injury [8,41,42,43]. Pioglitazone showed anti-inflammatory properties in diseases of the central nervous system such as stroke [44,45], Parkinson’s disease [46], and spinal cord injury [47]. More recently, it has been reported that PPAR-γ agonist affects microglia by reducing the pro-inflammatory M1 phenotype and by increasing the anti-inflammatory M2 phenotype after ischemic stroke [22,48,49].

## 4. Materials and Methods

### 4.1. Chemicals

Unless otherwise stated, all chemicals used in this study were purchased from Sigma (St. Louis, MO, USA). Monoclonal mouse antibodies against O4 and adenomatous polyposis coli (clone CC1, APC-CC1), and polyclonal rabbit antibodies against oligodendrocyte transcription factor 2 (Olig2) were purchased from Millipore (Billerica, MA, USA). Polyclonal rabbit antibodies against ionized calcium-binding adapter molecule 1 (Iba1) were obtained from Wako Chemicals USA (Irvine, CA, USA). Enzyme-linked immunosorbent assay (ELISA) kits for immunoassays of rat IL-1β, IL-6 and TNF-α were purchased from R&D Systems (Minneapolis, MN, USA).

### 4.2. Animals

Timed pregnant Sprague–Dawley rats arrived at the animal facility on day 19 of gestation. Animals were maintained in a room with a 12-h light/dark cycle and at constant temperature (22 ± 2 °C). The day of birth was defined as postnatal day 0 (P0). After birth, the litter size was adjusted to 12 pups per litter to minimize the effect of litter size on body weight and brain size. All procedures for animal care were conducted in accordance with the National Institutes of Health (NIH) Guide for the Care and Use of Laboratory Animals and were approved by the Institutional Animal Care and Use Committee at the University of Mississippi Medical Center and Fu Jen Catholic University, for animal protocol with number 1267B approved on 1 September 2016 and A10552 approved on Jan 4, 2017, respectively. Every effort was made to minimize the number of animals used and their suffering.

### 4.3. Animal Treatment

A total of 64 rats from eight litters were used in the present study. One pup from each litter was assigned to each group to obtain an n number of eight for each group while maintaining a male to female ratio of 1:1 in each group (4 males and 4 females). Pups were randomly divided into four groups: Saline + Vehicle; Saline + Pioglitazone; LPS + Vehicle; LPS + Pioglitazone. An intraperitoneal (i.p.) injection of LPS (2 mg/kg, from *Escherichia coli*, serotype 055:B5) was administered to 5-day-old (P5) Sprague–Dawley rat pups of both sexes. The control rats were injected with the same volume of sterile saline (0.1 mL). All animals survived the injection. Both LPS- and saline-injected animals were further divided into two groups: one received i.p. injections of a dose of 20 mg/kg pioglitazone dissolved in 20% DMSO in PBS and the other received the vehicle (20% DMSO/PBS) [50] within five minutes of the LPS injection. Core temperature was acquired using a rectal probe and digital thermometer (Fisher Scientific, Suwanee, GA, USA) every hour, beginning from just prior to LPS injection until 9 and 24 h after the LPS injection, with hypothermia defined as rectal temperature less than 32 °C [13]. Room temperature was maintained at 24 ± 2 °C throughout the study. Behavioral tests were conducted from P5 to P6. Following the behavioral tests, fresh brain tissue for mitochondrial complex I activity, ELISA, and lipid peroxidation assays was obtained via decapitation. Eight (4 males and 4 females) additional rats from each group were sacrificed by transcardiac perfusion with normal saline followed by fixation in 4% paraformaldehyde for brain sectioning preparation. Free-floating coronal brain sections 40 µm thick were collected for immunohistochemistry staining using a sliding microtome (Leica, SM 2010R, Wetzlar, Germany).

### 4.4. Behavioral Testing

Behavioral tests were performed as described previously with modifications [15,20,51]. The developmental test battery was based on previously well-established tests for neurobehavioral toxicity [14,52,53,54]. Hind-limb suspension, wire-hanging maneuver, negative geotaxis and righting reflex tests were performed on all rat pups at P6 as indicators of neurological function at early developmental stages. Body weights of rats were also recorded on P6. All animals were tested in the same order.

*Hind-Limb Suspension Test:* This test was used to evaluate the proximal hind-limb muscle strength, weakness and fatigue in rat neonates [14]. Each pup was given three trials on P6. In each trial, rat pups were placed head down with their hind limbs suspended by the lip of a plastic cylinder (4 cm inside diameter and 16 cm height). A cotton ball cushion was placed at the bottom to protect the animal’s head upon its fall. Suspension latencies were recorded, and the cut-off time was 120 s.

*Wire Hanging Maneuver:* This maneuver tests neuromuscular and locomotor development [53,54]. Pups suspended by their forelimbs from a horizontal rod (5 × 5 mm^2^ in cross-sectional area, 35-cm long, between two 50 cm-high poles) tend to support themselves with their hind limbs, preventing them from falling and aiding in progression along the rod. A sawdust-filled box at the base served as protection for the falling pups. Each pup was given three trials on P6 and suspension latencies were recorded. The cut-off time was 120 s.

*Negative Geotaxis:* This test is believed to test reflex development, motor skills and vestibular labyrinth and cerebellar integration [53,54]. Rats placed on a 15° incline with their head pointing down the slope turn to face upward and begin to crawl up the slope. Each pup was given three trials on P6 and the time required for a 180° turn upward was recorded. The cut-off time was 60 s.

*Righting Reflex:* Righting reflex is used as a test for reflection of muscle strength and subcortical maturation [53,54]. Pups were placed on their backs, and the time required to roll over onto all four feet on the platform was measured. Each pup was given three trials on P6 and the time spent for a rollover was recorded. The cut-off time was 60 s.

### 4.5. Immunohistochemistry

Brain injury was estimated based on the results of immunohistochemistry in consecutive brain sections prepared from rats sacrificed 1 day (P6) after LPS injection. For immunohistochemistry staining, primary antibodies were used with the following dilutions: O4, 1 μg/mL; APC-CC1, 1:200; Olig2, 1:500; or Iba1, 1:500. O4, APC-CC1 and Olig2 were used to detect late, mature OL progenitor cells, and total oligodendrocytes in the white matter, respectively. Microglia were detected using Iba1 immunostaining, which recognizes both the resting and activated microglia. Sections were incubated with primary antibodies at 4 °C overnight and further incubated with secondary antibodies conjugated with fluorescent dyes (Alexa Fluor 488, 1:300, or Alexa Fluor 555, 1:500, Invitrogen, Carlsbad, CA, USA) for 1 h in the dark at room temperature. 4′,6-Diamidino-2-phenylindole (DAPI) (100 ng/mL) was used simultaneously to identify nuclei in the final visualization. Sections incubated in the absence of primary antibody were used as negative controls. The resulting sections were examined under a fluorescent microscope (Nikon Ni-E, Melville, NY, USA) at appropriate wavelengths.

### 4.6. Determination of Mitochondrial Complex I Activity

Complex I activity was determined using a spectrophotometric assay based on the quantification of the rate of oxidation of the complex I substrate nicotinamide adenine dinucleotide (NADH) to ubiquinone as described in previous studies [15,16] with modifications. Brain tissue from each pup was collected 24 h after LPS injection. The frozen brain tissue was homogenized mechanically, sonicated on ice in 10 mM Tris-HCl buffer (pH 7.2) containing 225 mM mannitol, 75 mM saccharose, and 0.1 mM ethylenediaminetetraacetic acid (EDTA), and then centrifuged (600× *g*) for 20 min at 4 °C to obtain postnuclear supernatants. The optical density of the supernatants (40 µg of sample protein in 1 mL assay mixture) was spectrophotometrically recorded at a wavelength of 340 nm for 200 s at 37 °C. The assay mixture was a potassium phosphate buffer (25 mM, pH 7.5) containing 2 mM potassium cyanide, 5 mM magnesium chloride, 2.5 mg/mL bovine serum albumin, 2 µM antimycin A, 100 µM decylubiquinone, and 300 μM NADH. The proportion of NADH oxidation sensitive to an excess of rotenone (10 µM) was attributed to complex I activity. This procedure minimizes the dissociation of rotenone from complex I by using small buffer volumes, low temperatures, and rapid analysis. The specific activity (nmol NADH oxidized/min/mg protein) of complex I (NADH-ubiquinone oxidoreductase) was calculated using a molar extinction coefficient ε_340nm_ = 6.22 mM^−1^ cm^−1^ [15,55]. Enzyme activities were expressed as nmol/min/mg of brain tissue. Complex I activity was calculated as follows: Complex I activity = [Rate (min^−1^)/ε_340nm_ (6.22 mM^−1^ cm^−1^)]/0.040 mg.

### 4.7. Measurement of Lipid Peroxidation

Lipid peroxidation was determined in brain samples by measuring malondialdehyde (MDA) levels as thiobarbituric acid-reactive substances (TBARS) [17,18,19]. Briefly, the brain tissues were homogenized in cell extraction buffer (~5 mg/200 µL) containing 2 µL of 5% BHT (butylated hydroxytoluene) on ice, and precipitated proteins were removed by centrifugation at 12,000× *g* for 10 min. 200 µL of brain supernatant were mixed with 300 µL 20% trichloroacetic acid (TCA) and incubated for 1 min. Then 300 µL 0.67% thiobarbituric acid (TBA) was added to the mixture. The reaction was heated to 100 °C for 60 min. After cooling, the mixture was centrifuged at 12,000× *g* for 10 min, and the absorbance of the supernatant was determined. The number of resulting pink-stained TBARS were determined spectrophotometrically at 532 nm in a 96-well plate reader (µQuant, Bio-Tek instruments Inc., Winooski, VT, USA). A calibration curve was generated using 1,1,3,3-tetramethoxypropane (malondialdehyde, MDA, ACROS Organic) as the standard, which were subjected to the same treatment as the samples. The results were expressed as nanomoles of TBARS (MDA equivalents) per milligram of protein (nmol MDA/mg protein).

### 4.8. Determination of IL-1β, IL1-6, TNF-α Protein Levels by ELISA

The level of proinflammatory cytokines IL-1β, IL1-6 and TNF-α were measured by ELISA, as previously described [20,56]. Brain tissue from each pup was collected 24 h after LPS injection. Previously, a study found that the LPS-stimulated increase in inflammatory cytokine IL-1β in the rat brain reaches a peak value at that time point [4]. Brain tissues were homogenized by sonication in 1 mL ice-cold PBS (pH 7.2) and centrifuged at 12,000× *g* for 20 min at 4 °C. The supernatant was collected, and the protein concentration was determined by the Bradford method. ELISA was performed according to the manufacturer’s instructions, and data were acquired using a 96-well plate reader (Bio-Tek instruments, Inc., Winooski, VT, USA). The cytokine contents were expressed as picogram of cytokines per milligram of protein.

### 4.9. Quantification of Data and Statistics

Our previous studies indicate that neonatal i.p. LPS injection induces white matter injury primarily in the cingulum of the forebrain [16,20]. Therefore, brain sections at the bregma level were used to assess all pathological changes caused by systemic LPS injection. Most immunostaining data were quantified by counting positively stained cells. When the cellular boundary was not distinct, numbers of DAPI-stained nuclei from the superimposed images were counted. Three digital microscopic images were randomly captured in each of three brain sections, and the number of positively stained cells in the three images was counted and averaged (cells/mm^2^). The mean value of cell counts from the three brain sections was used to represent each brain. For ease of comparison among the treatment groups, results were standardized as the average number of cells/mm^2^. In response to LPS challenge, the number of Iba1+ microglia increases, and the soma of these cells become larger. In addition to cell density, the Iba1 immunoreactivity was also quantified by calculating the percentage area of the whole image that contains Iba1 immunostaining using NIH software Image J [20].

## 5. Conclusions

Our current results show that neonatal systemic LPS exposure causes inflammatory responses, brain white matter injury and neurobehavioral disturbances including the loss of body weight, hypothermia, hind-limb suspension, wire-hanging maneuver, negative geotaxis, and righting reflex in neonatal rats. LPS exposure stimulated microglia activation, which resulted in the production of pro-inflammatory cytokines, ROS and related downstream molecules that further induce lipid peroxidation on cell membranes and mitochondria dysfunction. Moreover, LPS exposure resulted in downregulation of oligodendrocyte lineage cells, which may contribute to hypomyelination. Treatment with PPAR-γ agonist pioglitazone reduces LPS-induced loss of oligodendrocytes and mitochondrial activity, and sensorimotor neurobehavioral dysfunction associated with attenuation of LPS-induced increases of TBARS contents, IL-1β levels and the number of activated microglia in neonatal rats (Figure 10). These findings suggest that pioglitazone prevents neurobehavioral disturbances induced by systemic LPS exposure in neonatal rats, and that its neuroprotective effects are associated with its impact on microglial activation, IL-1β induction, lipid peroxidation, oligodendrocyte production and mitochondrial activity. Thus, PPAR-γ agonist pioglitazone may be useful for therapeutic effects in neonatal white matter disease with its properties of anti-inflammation, anti-oxidation, promoting of oligodendrocyte progenitor cell differentiation, and mitochondrial cell survival signaling.

## Figures and Tables

**Figure 1 ijms-22-06306-f001:**
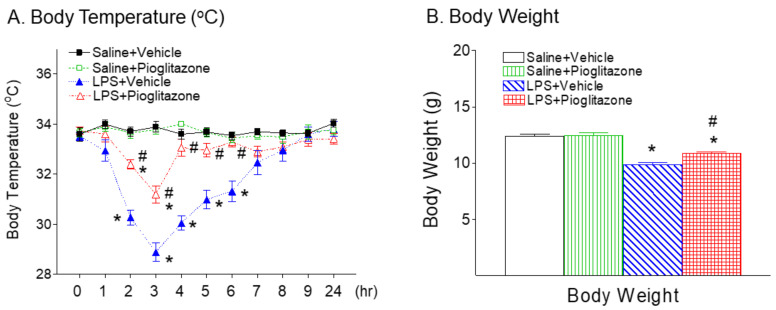
Pioglitazone attenuated neonatal LPS-induced hypothermia (**A**) and body weight loss (**B**) in the rat. The results are expressed as the mean ± SEM of eight animals in each group and analyzed by two-way repeated measures ANOVA for data from tests conducted continuously at different hours after LPS injection (**A**), or two-way ANOVA (**B**), followed by the Student–Newman–Keuls test. * *p* < 0.05 represents a significant difference for the LPS + Vehicle group or LPS + Pioglitazone group as compared with the Saline + Vehicle group. ^#^
*p* < 0.05 represents a significant difference for the LPS + Pioglitazone group as compared with the LPS + Vehicle group.

**Figure 2 ijms-22-06306-f002:**
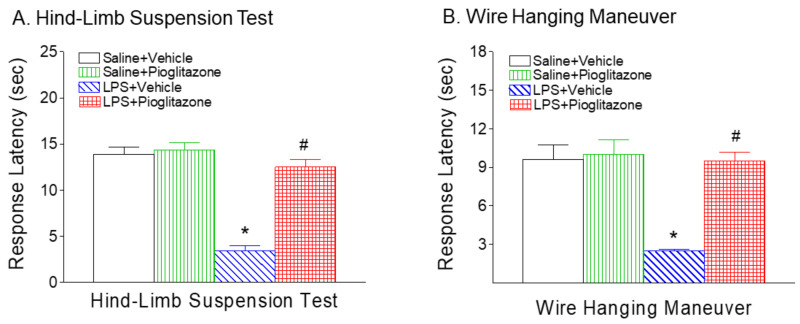
Pioglitazone attenuated neonatal LPS-induced reduction of mean latency times in hanging onto the edge of the tube in the hind-limb suspension test (**A**), and wire hanging maneuver (**B**) in the rat. The results are expressed as the mean ± SEM of eight animals in each group, and analyzed by two-way ANOVA, followed by the Student–Newman–Keuls test. * *p* < 0.05 represents a significant difference for the LPS + Vehicle group as compared with the Saline + Vehicle group. ^#^
*p* < 0.05 represents a significant difference for the LPS + Pioglitazone group as compared with the LPS + Vehicle group.

**Figure 3 ijms-22-06306-f003:**
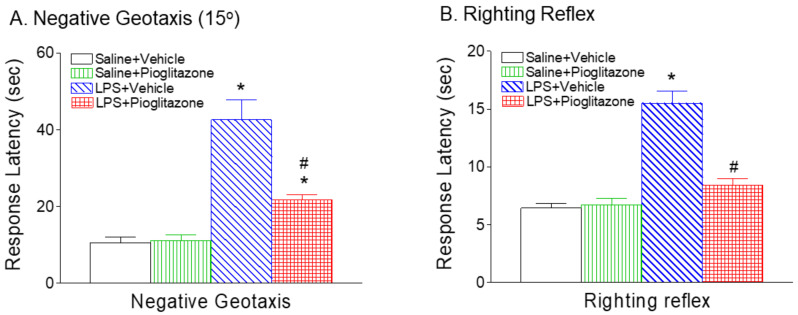
Pioglitazone attenuated neonatal LPS-induced elongation of mean latency times in negative geotaxis (**A**) and righting reflex (**B**) in the rat. The results are expressed as the mean ± SEM of eight animals in each group, and analyzed by two-way ANOVA, followed by the Student–Newman–Keuls test. * *p* < 0.05 represents a significant difference for the LPS + Vehicle group or LPS + Pioglitazone group as compared with the Saline + Vehicle group. ^#^
*p* < 0.05 represents a significant difference for the LPS + Pioglitazone group as compared with the LPS + Vehicle group.

**Figure 4 ijms-22-06306-f004:**
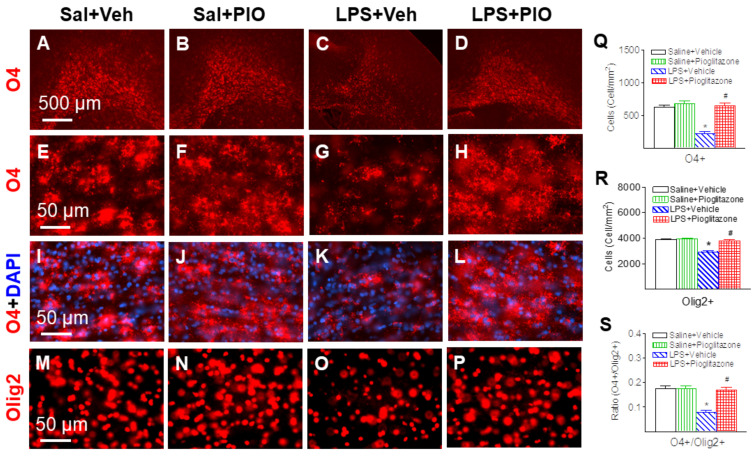
Pioglitazone attenuated neonatal LPS-induced reduction of late oligodendrocyte progenitor cells (O4+/O1−) and total oligodendrocytes (Olig2+) in the P6 rat brain. Abundant O4+ (Saline+Vehicle group: (**A**,**E**,**I**); Saline+Pioglitazone group: (**B**,**F**,**J**)), which positively stained the cell membrane and processes, and Olig2+ (Saline+Vehicle group: (**M**); Saline+Pioglitazone group: (**N**)) were found in the cingulum white matter of the brain sections at the bregma level in P6 Saline+Vehicle and Saline+Pioglitazone groups. DAPI (blue) was used simultaneously to identify nuclei in the final visualization (**I**–**L**). LPS injection reduced the number of normal O4+ cells (**C**,**G**,**K**) and Olig2+ cells (**O**). Pioglitazone attenuated LPS-induced loss of O4+ cells (**D**,**H**,**L**) and Olig2+ cells (**P**). Late oligodendrocyte progenitor cells and total oligodendrocytes were quantified by counting the number of O4+ cells (**Q**) and Olig+ cells in the cingulum white matter of the brain (**R**), respectively. The ratio of late oligodendrocyte progenitor cells (O4+/O1−) to total oligodendrocytes (Olig2+) in each group were shown in (**S**). The results are expressed as the mean ± SEM of eight animals in each group, and analyzed by two-way ANOVA, followed by the Student–Newman–Keuls test. * *p* < 0.05 represents a significant difference for the LPS + Vehicle group as compared with the Saline + Vehicle group. ^#^
*p* < 0.05 represents a significant difference for the LPS + Pioglitazone group as compared with the LPS + Vehicle group.

**Figure 5 ijms-22-06306-f005:**
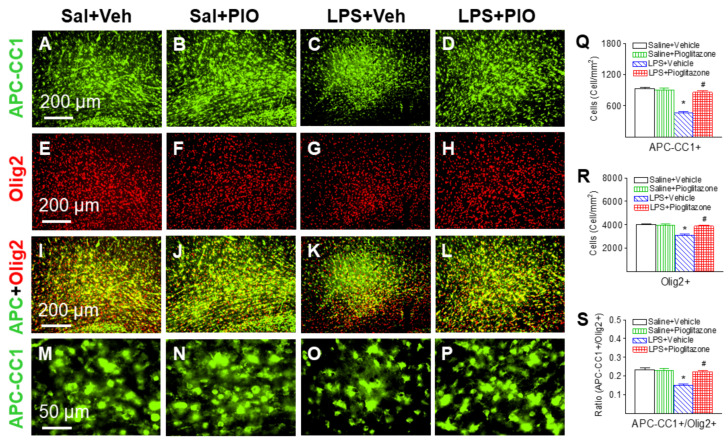
Pioglitazone attenuated neonatal LPS-induced reduction of mature oligodendrocyte cells (APC-CC1+, green) and total oligodendrocytes (Olig2+, red) in the P6 rat brain. APC-CC1+ oligodendrocytes (Saline+Vehicle group: (**A**,**I**,**M**); Saline+Pioglitazone group: (**B**,**J**,**N**)) and Olig2+ (Saline+Vehicle group: (**E**,**I**); Saline+Pioglitazone group: (**F**,**J**)) were found in the cingulum white matter of the brain sections at the bregma level in P6 Saline+Vehicle and Saline+Pioglitazone groups. Double-labeling (yellow) showed that APC-CC1-positive cells were Olig2+ cells (**I**–**L**). (**I**–**L**) are merged images of (**A**–**D**) and (**E**–**H**). LPS injection reduced the number of APC-CC1+ cells (**C**,**K**,**O**) and Olig2+ cells (**G**,**K**). Pioglitazone attenuated LPS-induced loss of APC-CC1+ cells (**D**,**L**,**P**) and Olig2+ cells (**H**,**L**). Mature oligodendrocyte cells and total oligodendrocytes were quantified by counting the number of APC-CC1+ cells (**Q**) and Olig2+ cells (**R**) in the cingulum white matter of the brain. The ratio of mature oligodendrocytes (APC-CC1+) to total oligodendrocytes (Olig2+) in each group were shown in (**S**). The results are expressed as the mean ± SEM of eight animals in each group, and analyzed by two-way ANOVA, followed by the Student–Newman–Keuls test. * *p* < 0.05 represents a significant difference for the LPS + Vehicle group as compared with the Saline + Vehicle group. ^#^
*p* < 0.05 represents a significant difference for the LPS + Pioglitazone group as compared with the LPS + Vehicle group.

**Figure 6 ijms-22-06306-f006:**
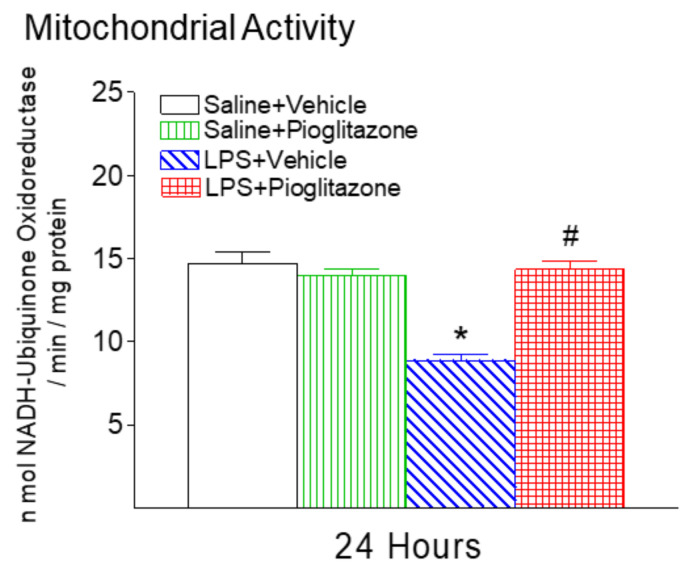
Pioglitazone attenuated neonatal LPS-induced reduction of mitochondrial complex I enzymatic activity in the rat brain. The results are expressed as the mean ± SEM of eight animals in each group, and analyzed by two-way ANOVA, followed by the Student–Newman–Keuls test. * *p* < 0.05 represents a significant difference for the LPS + Vehicle group as compared with the Saline + Vehicle group. ^#^
*p* < 0.05 represents a significant difference for the LPS + Pioglitazone group as compared with the LPS + Vehicle group.

**Figure 7 ijms-22-06306-f007:**
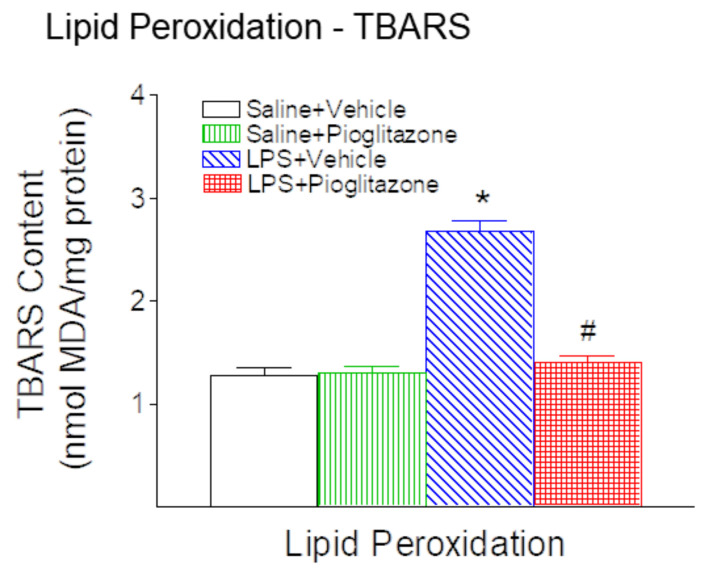
Pioglitazone reduced neonatal LPS exposure-induced increases in thiobarbituric acid-reactive substances (TBARS) content in the rat brain 24 h (P6) after LPS injection. Following LPS injection, the TBARS content in the brain was elevated compared with the control group in the P6 rat. Pioglitazone reduced LPS-induced increases in the TBARS content in the P6 rat brain. The results are expressed as the mean ± SEM (malondialdehyde, MDA equivalents) of eight animals in each group and analyzed by two-way ANOVA, followed by the Student-Newman-Keuls test. * *p* < 0.05 represents a significant difference for the LPS + Vehicle group as compared with the Saline + Vehicle group. ^#^
*p* < 0.05 represents a significant difference for the LPS + Pioglitazone group as compared with the LPS + Vehicle group.

**Figure 8 ijms-22-06306-f008:**
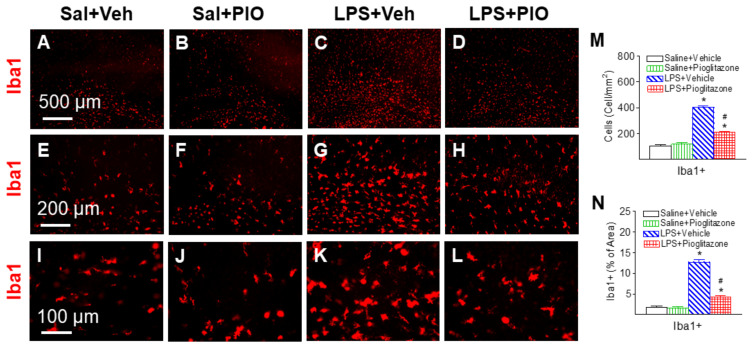
Pioglitazone attenuated neonatal LPS-induced microglia activation, as assessed by ionized calcium binding adapter molecule 1 (Iba1+) staining in the P6 rat brain. As shown by Iba1+ immunostaining in the cingulum white matter, a few microglia at the resting status with a small rod shaped soma and ramified processes were found in the Saline+Vehicle (**A**,**E**,**I**) and Saline+Pioglitazone (**B**,**F**,**J**) rat brains. LPS exposure induced the increase of numerous activated microglia with enlarged cell bodies and blunt processes (**C**,**G**,**K**). Pioglitazone attenuated an LPS-induced increase in activated microglia (**D**,**H**,**L**). Quantitation of the number of Iba1+ cells (**M**) and the percentage area of image that contained Iba1 staining (**N**) in the rat brain was performed as described in Methods. The results are expressed as the mean ± SEM of eight animals in each group, and analyzed by two-way ANOVA, followed by the Student–Newman–Keuls test. * *p* < 0.05 represents a significant difference for the LPS + Vehicle group or LPS + Pioglitazone group as compared with the Saline + Vehicle group. ^#^
*p* < 0.05 represents a significant difference for the LPS + Pioglitazone group as compared with the LPS + Vehicle group.

**Figure 9 ijms-22-06306-f009:**
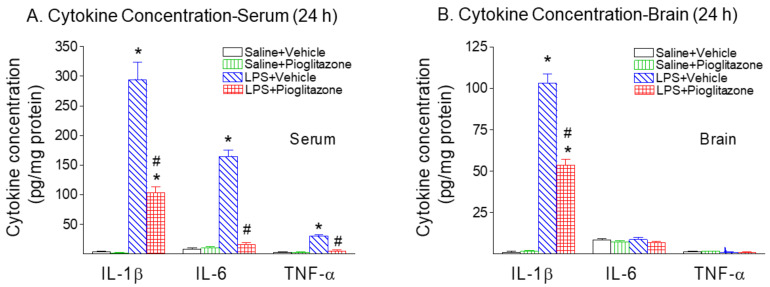
Pioglitazone attenuated neonatal LPS-stimulated increases in inflammatory cytokines (IL-1β, IL-6 and TNF-α) in the rat serum (**A**) and brain (**B**) 24 h after LPS injection. IL-1β, IL-6 and TNF-α concentrations were determined by ELISA and presented as units of pg/mg protein, as described in the Methods section. Twenty-four hours following LPS injection, serum levels of IL-1β, IL-6 and TNF-α (**A**), and brain levels of IL-1β were elevated as compared with the Saline+Vehicle group. Treatment with Pioglitazone attenuated LPS-induced contents of IL-1β, IL-6 and TNF-α. The results are expressed as the mean ± SEM of eight animals in each group, and analyzed by two-way ANOVA, followed by the Student–Newman–Keuls test. * *p* < 0.05 represents a significant difference for the LPS + Vehicle group or LPS + Pioglitazone group as compared with the Saline + Vehicle group. ^#^
*p* < 0.05 represents a significant difference for the LPS + Pioglitazone group as compared with the LPS + Vehicle group.

**Figure 10 ijms-22-06306-f010:**
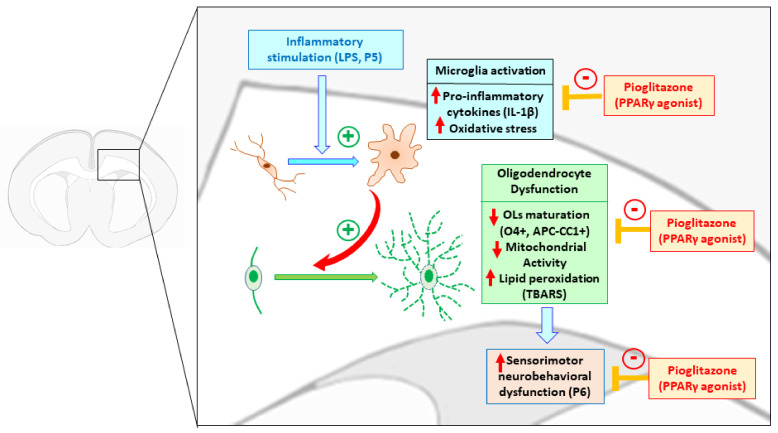
Schematic illustration of microglia-oligodendrocytes interaction in the brain white matter in neonatal systemic inflammation-induced sensorimotor neurobehavioral dysfunction. Systemic lipopolysaccharide (LPS) exposure led to stimulation of microglia activation, induction of pro-inflammatory cytokine interleukin-1β (IL-1β), loss of oligodendrocyte lineage cells (O4+/APC-CC1+) and mitochondria activity, and upregulation of lipid peroxidation which caused brain white matter injury and related sensorimotor neurobehavioral dysfunction in neonatal rats. Treatment with peroxisome proliferator-activated receptor gamma (PPARγ) agonist pioglitazone reduces LPS-induced loss of oligodendrocytes and mitochondrial activity, and sensorimotor neurobehavioral dysfunction, which was associated with attenuation of LPS-induced increment of thiobarbituric acid reactive substances (TBARS) contents, IL-1β levels and number of activated microglia in neonatal rats. These findings suggest that PPAR-γ agonist pioglitazone may be useful for therapeutic effects in neonatal white matter disease with its properties of anti-inflammation, anti-oxidation, promoting of oligodendrocyte progenitor cell differentiation, and mitochondrial cell survival signaling.

## Data Availability

The data presented in this study are available on request from the corresponding author. The data are available following Institutional and Federal guidelines requesting the use of a data-sharing agreement to impose appropriate limitations on users, to ensure data security at the recipient site and to prevent data manipulation.

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
