# Peer review of "Pioglitazone Ameliorates Lipopolysaccharide-Induced Behavioral Impairment, Brain Inflammation, White Matter Injury and Mitochondrial Dysfunction in Neonatal Rats"

_ijms, 2021, doi:10.3390/ijms22126306_

Round 1

Reviewer 1 Report

In this manuscript, authors investigated the effects of pioglitazone on LPS-induced behavioral impairment, brain inflammation, white matter injury and mitochondrial dysfunction in neonatal rats. Authors showed interesting data, however, there are several concerns as below.

major concerns

LPS-injected (at P5) rats shows significantly lower body weight (around 20% smaller than control rats) at P6, meaning that there could be severe dehydration (and malnutrition) due to acute inflammation or septic shock. Authors evaluated hind-limb suspension test, wire hanging maneuver, negative geotaxis and righting reflex as neurobehavioral deficits. However, such severe dehydration can greatly affect these results, and in such conditions these tests may not reflect neurobehavioral deficits correctly. Since previous researchers including authors themselves have reported that neonatal LPS injection had long-lasting effects on glial cells, authors should conduct these neurobehavioral evaluation on later time point when they recover from acute inflammation/dehydration.

Similarly, the reduction of late oligodendrocyte progenitor cells / mature oligodendrocyte, microglia activation, mitochondrial activity or lipid peroxidation at later time point should be evaluated .

minor comments

L6, “Lu-Tai tien and” might be “and Lu-Tai Tien”.

L107, “Similar to our previous study [15, 21])” should be “Similar to our previous study [15, 21]”.

L201, “p <0.05” should be “p < 0.05”

Author Response

-Reviewer 1

Q1: LPS-injected (at P5) rats shows significantly lower body weight (around 20% smaller than control rats) at P6, meaning that there could be severe dehydration (and malnutrition) due to acute inflammation or septic shock. Authors evaluated hind-limb suspension test, wire hanging maneuver, negative geotaxis and righting reflex as neurobehavioral deficits. However, such severe dehydration can greatly affect these results, and in such conditions, these tests may not reflect neurobehavioral deficits correctly. Since previous researchers, including the authors, have reported that neonatal LPS injection had long-lasting effects on glial cells; authors should conduct these neurobehavioral evaluations on later time point when they recover from acute inflammation/dehydration.

Response: We appreciate the reviewer for the comprehensive review of our current manuscript and its relation to our previous publications. The possible contributions of dehydration to the poor neurobehavioral performance in the LPS group should not be excluded. We have revised the discussion to include these comments. (pages 9-10)

Q2: Similarly, the reduction of late oligodendrocyte progenitor cells / mature oligodendrocyte, microglia activation, mitochondrial activity or lipid peroxidation at later time point should be evaluated.

Response: We appreciate reviewer’s suggestion. Our additional experiments are ongoing, and our goal is to determine the protective effects of pioglitazone on LPS-induced brain injury at a later time point (P21). The long-lasting protective effects of pioglitazone will be reported in the future.

Q3: minor comments

L6, “Lu-Tai tien and” might be “and Lu-Tai Tien”.

L107, “Similar to our previous study [15, 21])” should be “Similar to our previous study [15, 21]”.

L201, “p <0.05” should be “p < 0.05”

Response: We have revised all minor questions as reviewer suggested.

Reviewer 2 Report

This Manuscript discuss the impact of Pioglitazone on glia cells, especially micro and oligoglia. Including in vivo and biochemical studies, proved positive impact of drug to acute effects of LPS-triggered inflammation. Please, find below a list of my concerns regarding this Manuscript.

  1. The microphotographs are bit unfocused. Of course, I am aware that the dense cellular markers will give significant background at image, but this time I cannot distinguish even a couple of positive cells.
  2. Please include the low magnification images of brain tissues, which will help to establish if reported changes have a local or rather general input on brain tissue.
  3. Images showed at Fig. 4 suggest possible differences between control and Pioglitazone treatment, while images from Fig. 5, in my opinion, do not fallow the data showed in neighbor graph.
  4. Please, explain in details, how the calculations of fluorescence image have been done. Assuming, axis OY represent the number of DAPI (+) cells, while OX the intensity of fluorescence per image. In that case, I have some concerns, the DAPI staining will represent of cell kinds captured under the microscope, while the fluorescence data reports only marker positive staining. In that case, the final results will depend from the composition of exanimated tissue. Thus, please present the data only as a total fluorescence per image and include data from control staining. As control staining, please stain brain tissue with 2 different markers (marker of interest, for oligodentrocyts- O4 or progenitor marker and for microglia -Mac2 inflammation marker and basic marker, for oligodentrocytes CNPase and for microglia Iba1 basic marker) in 2 different channels. Then report the ratio between fluorescence intensity (marker of interest/basic marker) taken from the same brain tissue.
  5. In my opinion, the discussion is giving strong conclusions based on not as strong evidence, e.g. changes in “mitochondrial activity” (conclude from COX I activity assay only), hypnotized further hypomyelination (based only on progenitor and O4 staining made from acute experiments). I do understand the overall idea of the Manuscript, but I think that we should play with what we have and should be more careful in order to not mislead the future Readers.
  6. Important to note, the Authors represent very nice panel of behavioral studies, although the outcomes have not been linked further with biochemical studies.

Author Response

Q1: The microphotographs are bit unfocused. Of course, I am aware that the dense cellular markers will give significant background at image, but this time I cannot distinguish even a couple of positive cells.

Response: We have provided more images and methods for Figure 4 (page 5), Figure 5 (page 6) and Figure 8 (page 8).

Q2: Please include the low magnification images of brain tissues, which will help to establish if reported changes have a local or rather general input on brain tissue.

Response: We have added the low magnification images of brain tissues for Figure 4 (page 5), Figure 5 (page 6) and Figure 8 (page 8) according to the reviewer's suggestions.

Q3: Images showed at Fig. 4 suggest possible differences between control and Pioglitazone treatment, while images from Fig. 5, in my opinion, do not fallow the data showed in neighbor graph.

Response: We have added more images in Figure 4 (page 5), Figure 5 (page 6).

Q4: Please, explain in details, how the calculations of fluorescence image have been done. Assuming, axis OY represent the number of DAPI (+) cells, while OX the intensity of fluorescence per image. In that case, I have some concerns, the DAPI staining will represent of cell kinds captured under the microscope, while the fluorescence data reports only marker positive staining. In that case, the final results will depend from the composition of exanimated tissue. Thus, please present the data only as a total fluorescence per image and include data from control staining. As control staining, please stain brain tissue with 2 different markers (marker of interest, for oligodentrocyts- O4 or progenitor marker and for microglia -Mac2 inflammation marker and basic marker, for oligodentrocytes CNPase and for microglia Iba1 basic marker) in 2 different channels. Then report the ratio between fluorescence intensity (marker of interest/basic marker) taken from the same brain tissue.

Response: Abundant late oligodendrocyte progenitor cells (O4+/O1–), which had positive staining in the cell membrane and processes. We have added the description that DAPI was used simultaneously to identify nuclei in the final visualization (blue) (Figures 4I-4L) (page 4). We added the double labeling of mature oligodendrocytes (APC-CC1+) and total oligodendrocytes (Olig2+), and the ratio of mature oligodendrocytes (APC-CC1+) to total oligodendrocytes (Olig2+) in each group (Figure 5S) (page 6). In response to LPS challenge, not only the number of Iba1+ microglia increases, but also the soma of these cells become larger. Therefore, Iba1 staining was quantified by measuring both the number of Iba1+ cells and the percentage area that contains Iba1 immunostaining in the captured images Figure 8 (page 8). We plan to stain brain tissue with two different markers for microglia in future studies.

Q5: In my opinion, the discussion is giving strong conclusions based on not as strong evidence, e.g. changes in “mitochondrial activity” (conclude from COX I activity assay only), hypnotized further hypomyelination (based only on progenitor and O4 staining made from acute experiments). I do understand the overall idea of the Manuscript, but I think that we should play with what we have and should be more careful in order to not mislead the future Readers.

Response: We agree with the reviewer’s comments. We have added that suggestion further studies are needed. (page 11)

Q6: Important to note, the Authors represent very nice panel of behavioral studies, although the outcomes have not been linked further with biochemical studies.

Response: We appreciate the reviewer’s comments. We have revised the discussion to include a link between the behavioral and biochemical studies. (page 10)

Reviewer 3 Report

A very very good paper on the pioglitazone rescuing effects on the LPS-induced behavioral , inflammatory, WMI and mitochondrial deficits in neonatal rats. I can only congratulate the authors on such a good work. Introduction and Discussion sections are specifically summarizing and discussing the current literature on this matter. Methodology is complex and well described. No issue on the results and statistical part. Since this work implies the usage of mammals in experiments I would kindly ask the authors to provide the number of the ethical code in their manuscript. I appreciate they stated that efforts were made to reduce the suffering of the animals involved. After solving this, I think the paper can be accepted, as it has some interest for the readers of the journal.

Author Response

Q1: Since this work implies the usage of mammals in experiments, I would kindly ask the authors to provide the number of the ethical code in their manuscript.

Response: We appreciate the reviewer’s comments and we have added the “animal protocol with number 1267B approved on September 1, 2016 and A10552 approved on Jan 4, 2017,” in the 4.2. Animals in the methods. (page 12)

Round 2

Reviewer 1 Report

In this revised manuscript, authors responded to reviewers’ comments and the manuscript is improved. I believe it can be accepted. Thank you for authors’ work.

Reviewer 2 Report

No further comments.